# CARE: Causal Intervention and Adversarial Learning for Robust Multimodal Intent Recognition

## Abstract

Multimodal Intent Recognition (MIR) plays a key role in advancing human-computer interaction, yet its reliability is often challenged by spurious correlations and missing modalities in real-world data. Existing approaches, which mainly rely on complex fusion architectures or contrastive alignment, generally do not account for the underlying causal structures of multimodal signals, resulting in limited generalization and robustness. They typically treat missing modalities as a data issue addressed by passive imputation rather than an opportunity to learn deeper, causally-informed representations. To address these limitations, we propose the Counterfactual Adversarial Representation Enhancement (CARE) framework, which reframes MIR as a causal learning problem. CARE implements causal principles through two complementary modules: a counterfactual generation module that interprets modality completion as a causal intervention to capture shared, abstract concepts across modalities, and an adversarial de-confounding mechanism. The latter employs a Gradient Reversal Layer and a modality discriminator to remove the confounding effects of the modality combination, enforcing the learning of intervention-invariant representations. This dual approach ensures that the learned intent features are both robust to missing data and causally consistent. We evaluate CARE extensively on the MIntRec and the more challenging MIntRec2.0 datasets. Results show that CARE achieves state-of-the-art performance, surpassing the strongest baseline by up to 4.41% in WF1 and 12.03% in recall, while maintaining high robustness under various missing-modality scenarios. This work introduces a principled paradigm for building causally robust multimodal systems, providing a systematic way to mitigate confounding bias and improve generalization in complex, real-world interactive environments.

## 1 Introduction

Integrating multimodal signals such as text, audio, and video to accurately interpret human intentions is critical to advancing human-computer interaction and enabling effective human-machine collaboration. Multimodal Intent Recognition (MIR) is central to this endeavor, directly influencing system intelligence and the quality of user interactions in applications like dialogue systems and assistive robots.

A core challenge in MIR is disentangling genuine causal features of intent from spurious correlations introduced by confounders. Modalities are often entangled: text may convey fear or anger, while an animated audio tone and a visual smirk suggest a different emotional state (Figure 1). Models that rely solely on such correlations can easily misinterpret user intent, motivating a necessary shift from associative information fusion toward robust causal reasoning. While prior work has focused on sophisticated fusion architectures and contrastive alignment, these methods often fail to capture underlying causal structures or handle missing modalities effectively beyond simple imputation.

To address these limitations, we adopt a causal perspective and propose the CARE (**C**ounterfactual **A**dversarial **R**epresentation **E**nhancement) framework, which consists of two interacting modules. The first module reframes modality completion as a self-supervised counterfactual generation task. When a modality is missing, the framework generates a plausible substitute, compelling the model

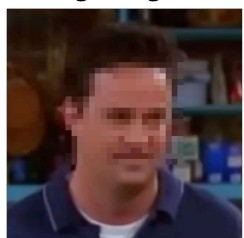

***The Big Bang Theory***

Chandler: Oh my god! You almost gave me a heart attack!

- **Text:** suggests fear or anger.
- **Audio:** animated tone
- **Video:** smirk, no sign of anxiety

Figure 1: Example of multimodal incongruence: text conveys fear or anger, audio has an animated tone, and video shows a smirk without anxiety. Such conflicting cues may mislead models into inferring incorrect intent. The image is adapted from a TV sitcom ("The Big Bang Theory") and has been pixelated to avoid copyright conflicts.

to learn representations that transcend simple correlations. The second module employs an adversarial debiasing mechanism that treats the source of a feature—whether observed or counterfactually generated—as a proxy confounder. By using a Gradient Reversal Layer (GRL) and a modality discriminator, our feature extractor is trained to produce intent representations that are invariant to this confounder, making them more causally robust. The contributions of this work are threefold:

i. We introduce a causal learning framework that systematically generates counterfactual representations and integrates them into downstream training to capture robust cross-modal causal relationships.

ii. We design an adversarial debiasing mechanism that removes the confounding influence of the feature source (i.e., observed vs. generated), yielding representations that are invariant to such interventions and thus more reliable for decision-making.

iii. Extensive experiments on MIntRec and MIntRec2.0 demonstrate CARE's effectiveness. On MIntRec, CARE achieves accuracy (ACC %), weighted F1 (WF1 %), weighted precision (WP %), and recall (R %) of **76.63**, **76.56**, **77.28**, and **74.89**, surpassing baselines by **1.91**, **1.95**, **2.21**, and **2.95**. On MIntRec2.0, CARE reaches **59.86**, **59.46**, **59.59**, and **54.06**, outperforming state-of-the-art methods by **2.06**, **4.41**, **3.77**, and **12.03**.

## 2 RELATED WORK

This section reviews two research areas relevant to our study: MIR and causal inference in machine learning. We summarize progress in these domains and identify remaining challenges that motivate our framework.

### 2.1 MULTIMODAL INTENT RECOGNITION

MIR integrates text, audio, and video to infer user intentions, supporting natural human–computer interaction (Rossiter, 2011; Rysbek et al., 2023; Zhao et al., 2024). Applications include dialog systems (Wu et al., 2019), robotic collaboration (Dermy et al., 2017; Trick et al., 2019), autonomous driving (Okur et al., 2019), AR-based chemistry (Xia et al., 2023), and customer service (Yu et al., 2021). Benchmark datasets such as MIntRec (Zhang et al., 2022) and MIntRec2.0 (Zhang et al., 2024) have standardized evaluation protocols and accelerated research.

Early studies explored multimodal fusion, including combining speech and gesture (Paul et al., 2022), applying spatiotemporal GCNs for skeletal intent recognition (Shi et al., 2023), and integrating 3D CNNs with LSTMs for motion understanding (Wen & Wang, 2021). The introduction of the Transformer brought a major advance: MulT (Tsai et al., 2019) showed the value of cross-modal attention, inspiring research on multimodal extensions of large language models (Rahman et al., 2020) and adaptive fusion designs (Hu et al., 2025). More recently, contrastive learning has driven further progress. Examples include MIRCL (Wu et al., 2023), which strengthens feature discrimination, CAGC (Sun et al., 2024), which captures contextual dependencies, and TCL-MAP (Zhou

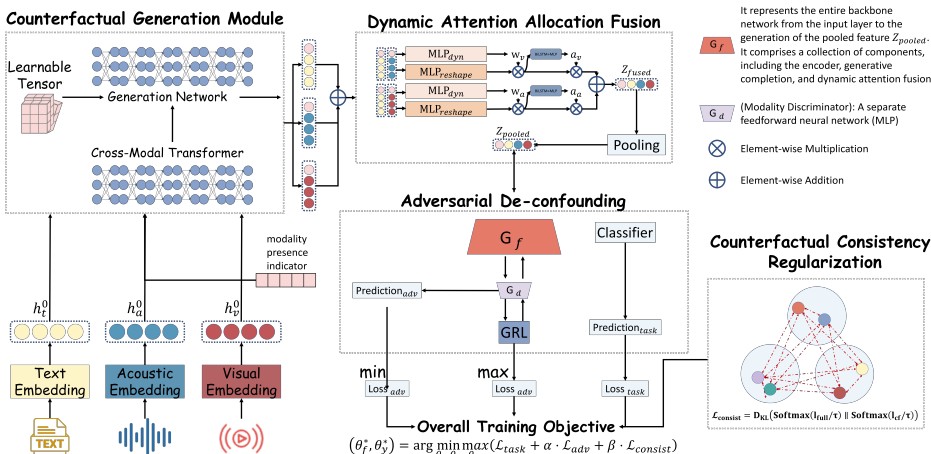

Figure 2: Overview of the CARE framework. Parallel encoders extract unimodal features, while the Counterfactual Generation module synthesizes missing modalities. Features are fused by the DAF module, and an Adversarial De-confounding stage enforces invariance to modality combinations before intent classification.

et al., 2024), which improves alignment through modality-aware prompts. Despite these advances, many methods remain sensitive to spurious correlations (Nguyen et al., 2024; Rysbek et al., 2023), limiting robustness in unseen conditions.

## 2.2 CAUSAL INFERENCE IN MACHINE LEARNING

Causal inference provides tools for distinguishing true causal effects from correlations and has supported more robust, fair, and interpretable learning (Yao et al., 2021; Rawal et al., 2025; Ma, 2024; Kosaraju, 2024). Techniques such as double/debiased ML, AIPW, and TMLE are valued for their double robustness in high-dimensional settings (Moccia et al., 2024; Kabata & Shintani, 2025). Classical instrumental variable methods mitigate unobserved confounding but are limited by the challenge of identifying valid instruments (Wu et al., 2025). Causal intervention, formalized through Pearl's *do*-calculus and counterfactual reasoning, enables systematic "what-if" analyses, isolating effects such as language versus visual input. Although its adoption in multimodal learning is still limited, prior work demonstrates its potential for improving generalization and reducing bias (Cho, 2024).

Research on MIR has shifted from fusion strategies to contrastive learning, while causal inference offers mechanisms for handling confounding. Their integration remains underexplored. To this end, we propose CARE, which combines counterfactual generation with adversarial de-confounding to learn intent representations that are both discriminative and causally robust.

## 3 METHOD

To address the challenges posed by confounding factors and spurious correlations in MIR, particularly under conditions of missing data, we introduce the CARE framework. The model is implemented as a structured pipeline that processes multimodal inputs to derive de-confounded representations for reliable intent prediction. The central idea is to operationalize causal intervention by generating counterfactual features that simulate "what-if" scenarios when modalities are absent. This mechanism encourages the model to learn modality-invariant representations.

The CARE pipeline, illustrated in Figure 2, sequentially performs: Parallel Modality Encoding with specialized encoders; Counterfactual Generation for missing modalities; Dynamic Attention Allocation Fusion (DAF) to integrate observed and generated features; and Adversarial De-confounding to enforce causal invariance, before a final Intent Prediction. The complete end-to-end training procedure is detailed in Algorithm 1 in Appendix A.1.

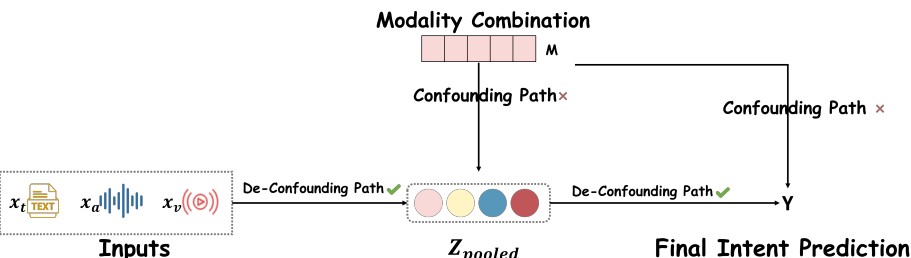

Figure 3: Causal graph. The modality indicator $M$ confounds both the representation $Z_{\text{pooled}}$ and the label $Y$. The objective is to eliminate the backdoor path by enforcing invariance to $M$.

## 3.1 NOTATION

The main mathematical symbols are defined in Table 1. The modality existence indicator $M \in \{0, \ldots, 6\}$, a discrete integer, encodes the seven possible combinations of the three modalities: text ($T$), audio ($A$), and video ($V$). The specific mapping for this encoding is detailed in Table 6 in Appendix A.2.

Table 1: Symbols and Definitions

| Symbol | Definition |
|---|---|
| $X_t, X_a, X_v$ | Raw input sequences of text, audio, and video. |
| $Y, \hat{Y}$ | Ground-truth and predicted intent labels. |
| $M$ | Discrete indicator of modality availability, $M \in \{0, \ldots, 6\}$, serving as a confounder in the causal model. |
| $G_f$ | Feature extractor including encoders, counterfactual generation, and Dynamic Attention Fusion (DAF). |
| $H_m^0$ | Initial encoded feature for modality $m \in \{t, a, v\}$. |
| $f_{\text{gen}}$ | Counterfactual generation operator guided by a learnable tensor. |
| $\tilde{H}_m$ | Feature after counterfactual generation; $\tilde{H}_m = H_m^0$ if observed, otherwise generated. |
| $Z_{\text{pooled}}$ | Unified multimodal representation from $G_f$, input to classifier and discriminator. |
| $G_d$ | Discriminator predicting modality combination $M$. |
| $G_y$ | Classifier for intent prediction. |
| $\mathcal{L}_{\text{task}}$ | Task loss (cross-entropy). |
| $\mathcal{L}_{\text{adv}}$ | Adversarial loss for de-confounding. |
| $\mathcal{L}_{\text{consist}}$ | Consistency loss defined by KL divergence between factual and counterfactual predictions. |
| $\alpha, \beta$ | Hyperparameters balancing the loss terms. |

## 3.2 PROBLEM FORMULATION

The task is MIR: given multimodal inputs $X$, the goal is to predict the intent label $Y$. A challenge arises because the specific modality combination may act as a confounder, creating spurious correlations. For example, visual cues may correlate with emotional labels in training data, causing the model to associate the presence of video with intent categories rather than semantic content. This introduces a confounding path $M \to Y$ that hinders generalization.

We therefore frame the task as a causal de-confounding problem. The modality combination $M$ is treated as the confounder. Figure 3 illustrates the causal graph, where $M$ affects both the representation $Z_{\text{pooled}}$ and the label $Y$, creating the backdoor path $Z_{\text{pooled}} \leftarrow M \to Y$. Our objective is to design a feature extractor $G_f$ that blocks this path.

Formally, we aim to ensure:

$$Y \perp M \mid Z_{\text{pooled}}. \tag{1}$$

To satisfy this, we adopt adversarial training (Section 3.6), where $G_d$ predicts $M$ while $G_f$ learns to produce $Z_{\text{pooled}}$ that conceals this information.

### 3.3 PARALLEL MODALITY ENCODERS

Each modality is encoded independently to produce unimodal features. The text input $X_t$ is processed by BERT (bert-large-uncased), a 24-layer Transformer with hidden size 1024, fine-tuned for intent recognition, producing the representation

$$H_t^0 = \text{BERT}(X_t). \tag{2}$$

For audio, the input $X_a$ is passed through a Bidirectional Peephole LSTM (Gers & Schmidhuber, 2000), which leverages cell-state connections to capture long-range temporal dependencies, yielding

$$H_a^0 = \text{BiPeepholeLSTM}(X_a). \tag{3}$$

The video sequence $X_v$ is encoded using a 6-layer Transformer encoder with 8 attention heads to model inter-frame relations, resulting in

$$H_v^0 = \text{TransformerEncoder}(X_v). \tag{4}$$

These unimodal features $\{H_t^0, H_a^0, H_v^0\}$ are then used as inputs for subsequent feature completion and fusion.

### 3.4 COUNTERFACTUAL GENERATION MODULE

To handle missing modalities, we frame feature synthesis as a causal intervention. The counterfactual generation operator, $f_{\text{gen}}$, produces a plausible substitute feature $\tilde{H}_{m'}$ for any missing modality $m'$. This operator consists of modality-specific projection networks followed by a final generation network, each implemented as lightweight modules with a 1D convolution of kernel size 1 and a LeakyReLU activation, functionally equivalent to a linear transformation at each sequence position. Synthesis begins by forming an aggregated representation, $H_{\text{agg}}$, created by concatenating a learnable, modality-specific generative seed tensor, $\text{Tensor}_{m'}$, with the outputs of projection networks that map the available modality features $\{H_m^0, m \in \mathcal{M}_{\text{avail}}\}$ to a shared dimension. In our work, the seed tensor has dimension $1024 \times 16$, providing a distinct initialization for each modality. For example, when text is missing, the aggregated representation is constructed as

$$H_{\text{agg}} = \text{concat}[\text{Tensor}_t, \text{Proj}_{a \to t}(H_a^0), \text{Proj}_{v \to t}(H_v^0)] \tag{5}$$

where $\text{Proj}_{a \to t}$ and $\text{Proj}_{v \to t}$ denote the audio-to-text and video-to-text projection networks, respectively. The aggregated representation is then processed by the generation network to produce the counterfactual feature $\tilde{H}_{m'}$, aligning with the target modality's sequence length and feature dimension. This design enables the model to dynamically synthesize contextually relevant features from the available modalities.

### 3.5 DYNAMIC ATTENTION ALLOCATION FUSION

The DAF module integrates $\{\tilde{H}_t, \tilde{H}_a, \tilde{H}_v\}$ into a unified representation. We condition the fusion on $\tilde{H}_t$ since text often provides explicit semantic cues.

For non-text modalities:

$$W_v = \sigma(\text{DynLayer}_v(\text{concat}(\tilde{H}_v, \tilde{H}_t))) \tag{6}$$

$$W_a = \sigma(\text{DynLayer}_a(\text{concat}(\tilde{H}_a, \tilde{H}_t))) \tag{7}$$

$$H_v' = W_v \odot \text{Linear}_v(\tilde{H}_v) \tag{8}$$

$$H_a' = W_a \odot \text{Linear}_a(\tilde{H}_a). \tag{9}$$

Attention vectors reweight these features:

$$\mathbf{a}_v = \text{Softmax}(\text{Linear}(\text{BiLSTM}(H_v'))) \tag{10}$$

$$\mathbf{a}_a = \text{Softmax}(\text{Linear}(\text{BiLSTM}(H_a'))). \tag{11}$$

The fused output is:

$$Z_{\text{fused}} = (\mathbf{a}_v \odot H_v') + (\mathbf{a}_a \odot H_a') + \tilde{H}_t, \tag{12}$$

followed by mean pooling to obtain $Z_{\text{pooled}}$.

The pooled representation $Z_{\text{pooled}}$ is fed into the intent classifier $G_y$, implemented as a linear layer followed by a softmax function. This classifier outputs a probability distribution over the intent classes. To train the model, we minimize the cross-entropy loss between the predicted distribution $\hat{Y}$ and the ground-truth one-hot label $Y$, which defines the task loss:

$$\mathcal{L}_{\text{task}} = \text{CrossEntropy}(\hat{Y}, Y) \tag{13}$$

### 3.6 ADVERSARIAL DE-CONFOUNDING

The adversarial de-confounding mechanism is realized through a GRL, following the framework of Domain-Adversarial Training of Neural Networks (Ganin et al., 2016). The GRL is placed between the feature extractor $G_f$ and the discriminator $G_d$. In the forward pass, it behaves as an identity mapping, while during backpropagation it multiplies the gradient by $-\lambda$ before passing it to $G_f$, thereby enforcing the adversarial objective. The discriminator $G_d$ attempts to infer the modality indicator $M$ from the pooled representation $Z_{\text{pooled}}$, and the adversarial loss is given by the cross-entropy between its prediction and the ground-truth indicator:

$$\mathcal{L}_{\text{adv}} = \text{CrossEntropy}\big(G_d(\text{GRL}\lambda(Z_{\text{pooled}})), M\big) \tag{14}$$

To avoid instability in training, we adopt a two-stage protocol in which adversarial learning is introduced gradually, ensuring that the optimization of the main task is not compromised.

### 3.7 COUNTERFACTUAL CONSISTENCY REGULARIZATION

To improve the model's predictive stability under various causal interventions, we incorporate a counterfactual consistency regularization applied to samples with all modalities present, denoted as $X_{\text{full}}$. For each complete input, the model's output logits are computed as $l_{\text{full}} = G_y(G_f(X_{\text{full}}))$. We then construct three counterfactual inputs, $\{X_{\text{cf},t}, X_{\text{cf},a}, X_{\text{cf},v}\}$, each simulating the absence of one modality (text, audio, or video), which are processed by the model—including the counterfactual generation step—to yield corresponding logits $\{l_{\text{cf},t}, l_{\text{cf},a}, l_{\text{cf},v}\}$. The consistency loss is defined as the average Kullback-Leibler (KL) divergence between the probability distributions of the complete and counterfactual logits, softened by a temperature parameter $\tau$:

$$\mathcal{L}_{\text{consist}} = \frac{1}{3} \sum_{m \in \{t,a,v\}} D_{\text{KL}}\Big(\text{Softmax}(l_{\text{full}}/\tau) \,\|\, \text{Softmax}(l_{\text{cf},m}/\tau)\Big). \tag{15}$$

This loss encourages the model to produce consistent predictions regardless of whether modality features are observed or generated, akin to consistency regularization in semi-supervised learning, thereby enhancing robustness against input perturbations (Bachman et al., 2014). In our experiments, the temperature is set to $\tau = 0.07$.

### 3.8 OVERALL TRAINING OBJECTIVE

The total loss is:

$$\mathcal{L}_{\text{total}} = \mathcal{L}_{\text{task}} + \alpha\mathcal{L}_{\text{adv}} + \beta\mathcal{L}_{\text{consist}}. \tag{16}$$

Our training methodology is organized in two sequential phases to achieve a balance between robustness and discriminative performance. In the first phase, robustness-oriented pre-training, the model learns generalizable representations that are resilient to missing modalities, facilitated by techniques such as modality dropout. Following this, the performance fine-tuning phase refines the model using a reduced learning rate, enabling improved task-specific discriminative capability. The complete details of this two-stage protocol are provided in Appendix A.3. AdamW is used for optimization, with $\alpha = 0.5$ and $\beta = 1.0$ selected by grid search. A detailed sensitivity analysis of these key hyperparameters is presented in Appendix B.1.

## 4 EXPERIMENTS

Details of the experimental setup, including hyperparameters and implementation choices, are provided in Appendix A.4.

## 4.1 DATASETS

We evaluate our framework on two MIR benchmarks: MIntRec (Zhang et al., 2022) and MIntRec2.0 (Zhang et al., 2024). Core statistics are listed in Table 2. MIntRec contains 2,224 samples from 43 dialogues with text, video, and audio modalities across 20 intent classes, serving as a standard reference for single-turn MIR.

MIntRec2.0 extends both scale and diversity, comprising 15,040 samples from 1,245 dialogues. It supports multi-turn, multi-party interactions and introduces 30 fine-grained intent classes. In addition, it includes more than 5,700 out-of-scope (OOS) samples, which pose a stronger challenge for intent recognition under unseen conditions.

These datasets enable evaluation not only of baseline MIR accuracy but also of robustness to complex dialogue structures and OOS cases. They further inform model development and provide evidence of generalization across varied conversational settings.

Table 2: Comparison of datasets used for MIR. Only key statistics relevant to model evaluation are shown.

| Dataset | Samples | Dialogues | Intent Classes | Modalities / OOS |
|---|---|---|---|---|
| MIntRec (Zhang et al., 2022) | 2,224 | 43 | 20 | Text, Video, Audio / No |
| MIntRec2.0 (Zhang et al., 2024) | 15,040 | 1,245 | 30 | Text, Video, Audio / Yes |

## 4.2 BASELINE METHODS

We compare CARE with four representative baselines in MIR.

**MulT** (Tsai et al., 2019) employs directional cross-modal attention to model interactions across unaligned sequences, addressing variable sampling rates and long-range dependencies. **MAG-BERT** (Rahman et al., 2020) incorporates nonverbal cues into pretrained transformers through a Multimodal Adaptation Gate, enhancing multimodal fine-tuning. **TCL-MAP** (Zhou et al., 2024) leverages token-level contrastive learning with modality-aware prompts to obtain discriminative intent representations. **MVCL-DAF** (Hu et al., 2025) integrates variational contrastive learning with dynamic attention allocation, weighting modalities adaptively by informativeness.

These baselines represent advances in cross-modal attention, pretrained adaptation, contrastive learning, and adaptive fusion, and together provide a solid reference for comparison.

## 4.3 MAIN RESULT

Table 3: Comparison of CARE against baseline models on the MIntRec and MIntRec2.0 datasets. Bold values denote the best performance, and underlined values denote the second-best performance. $\Delta$ indicates the absolute improvement of CARE over the strongest baseline.

| Method | MIntRec | | | | MIntRec2.0 | | | |
|---|---|---|---|---|---|---|---|---|
| | ACC | WF1 | WP | R | ACC | WF1 | WP | R |
| MulT (Tsai et al., 2019) | 72.52 | 71.80 | 72.60 | 67.44 | 56.95 | 54.26 | 54.49 | 40.65 |
| MAG-BERT (Rahman et al., 2020) | 72.16 | 71.30 | 72.03 | 67.61 | 55.87 | 52.58 | 53.71 | 39.93 |
| TCL-MAP (Zhou et al., 2024) | 73.69 | 73.38 | 73.90 | 71.59 | 56.99 | 54.33 | 55.07 | 41.87 |
| MVCL-DAF (Hu et al., 2025) | 74.72 | 74.61 | 75.07 | 71.94 | 57.80 | 55.05 | 55.82 | 42.03 |
| **our CARE** | **76.63** | **76.56** | **77.28** | **74.89** | **59.86** | **59.46** | **59.59** | **54.06** |
| $\Delta$ | 1.91↑ | 1.95↑ | 2.21↑ | 2.95↑ | 2.06↑ | 4.41↑ | 3.77↑ | 12.03↑ |

To assess the effectiveness of CARE, we compare it against four widely used multimodal baselines (MulT, MAG-BERT, TCL-MAP, and MVCL-DAF) on both MIntRec and MIntRec2.0. Quantitative results are presented in Table 3, and overall performance profiles are summarized in Figure 4.

Table 3 shows that CARE consistently surpasses all baselines. On MIntRec, it achieves the highest scores across all metrics, with absolute gains of up to 2.95% over the strongest competitor. On

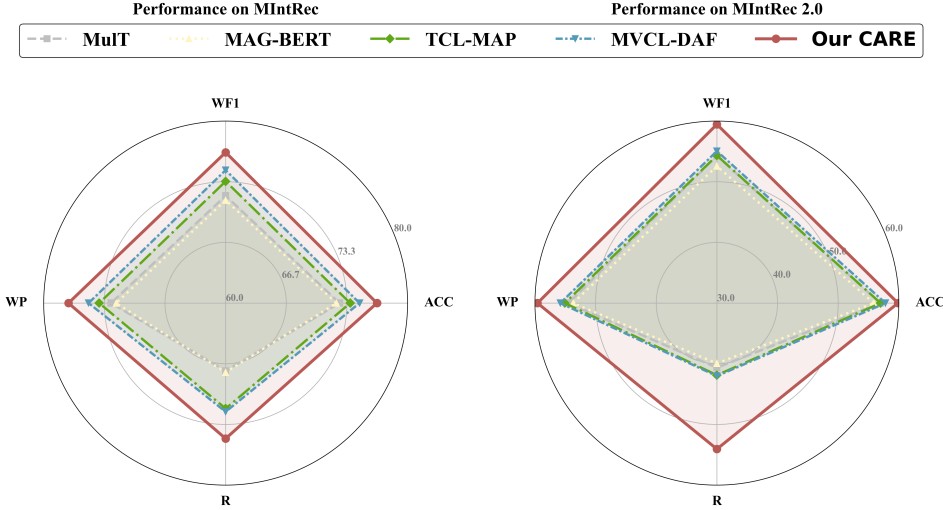

Figure 4: Radar visualization of model performance across four evaluation metrics (ACC, WF1, WP, and R) on the MIntRec (left) and MIntRec2.0 (right) datasets. CARE shows consistent improvements over all baselines across both benchmarks.

the more challenging MIntRec2.0 benchmark, CARE delivers a 12.03% increase in recall, together with notable improvements in accuracy, precision, and F1, indicating strong robustness and generalization. To complement these quantitative results, Appendix B.2 provides t-SNE visualizations demonstrating that CARE learns representations that separate intents clearly while remaining invariant to modality-specific noise.

The radar plots in Figure 4 reinforce this observation: CARE expands evenly across evaluation metrics, whereas baselines exhibit unbalanced profiles, particularly in recall and transfer to MIntRec2.0. Overall, the results indicate that CARE attains state-of-the-art performance while maintaining stable behavior across datasets.

## 4.4 ABLATION STUDY

We examine the contribution of CARE's main components by ablating the Counterfactual Generation, Adversarial De-confounding, and DAF modules. Results on MIntRec and MIntRec2.0 are reported in Table 4.

Each module plays an important role, though their effects vary in scale. The Counterfactual Generation module has the largest impact: removing it reduces accuracy by 7.52 points and weighted F1 by 9.00 points on MIntRec2.0, showing its effectiveness in handling scarce or missing modalities. The Adversarial De-confounding mechanism produces the next largest decline, with accuracy decreasing by 3.74 points and weighted F1 by 5.50 points, indicating that limiting spurious modality correlations improves robustness. Replacing DAF with simple pooling leads to smaller but consistent drops, reducing accuracy by 2.31 points and weighted F1 by 4.60 points, which supports the benefit of adaptive fusion. The sharper performance losses on MIntRec2.0 suggest that these components are especially valuable in more demanding conversational settings.

Table 4: Ablation study of the core components of our CARE framework on the MIntRec and MIntRec2.0 datasets. We report the performance (ACC, WF1, WP, R in %) of the full model and its variants. The values in parentheses indicate the performance drop ($\Delta \downarrow$) relative to the full CARE model, highlighting the contribution of each component.

| Model Variant | MIntRec | | | | MIntRec2.0 | | | |
|---|---|---|---|---|---|---|---|---|
| | ACC | WF1 | WP | R | ACC | WF1 | WP | R |
| **Full CARE** | **76.63** | **76.56** | **77.28** | **74.89** | **59.86** | **59.46** | **59.59** | **54.06** |
| w/o Counterfactual Generation | 72.51 (4.12↓) | 72.06 (4.50↓) | 72.85 (4.43↓) | 70.12 (4.77↓) | 52.34 (7.52↓) | 50.46 (9.00↓) | 51.98 (7.61↓) | 42.11 (11.95↓) |
| w/o Dynamic Attention Allocation Fusion | 74.65 (1.98↓) | 74.36 (2.20↓) | 75.01 (2.27↓) | 72.95 (1.94↓) | 57.55 (2.31↓) | 54.86 (4.60↓) | 55.43 (4.16↓) | 48.92 (5.14↓) |
| w/o Adversarial De-confounding | 74.18 (2.45↓) | 73.76 (2.80↓) | 74.53 (2.75↓) | 71.87 (3.02↓) | 56.12 (3.74↓) | 53.96 (5.50↓) | 54.88 (4.71↓) | 46.57 (7.49↓) |

## 4.5 CROSS-SCENARIO ROBUSTNESS UNDER INCOMPLETE MODALITIES

We assess whether combining causal intervention with adversarial learning enhances robustness under incomplete data by conducting cross-scenario evaluations against MVCL-DAF across seven modality configurations, as summarized in Table 5.

With full inputs (V+A+T), CARE consistently outperforms the baseline. On MIntRec2.0, it reaches 59.86% accuracy, a 2.06-point improvement over MVCL-DAF, showing that the proposed causal deconfounding yields cleaner and more discriminative representations when information is complete.

The advantage becomes more evident under missing-modality conditions. When text is absent (V+A), CARE raises accuracy on MIntRec2.0 from 46.91% to 48.33%. These results suggest that treating modality completion as causal intervention is effective: instead of simple imputation, the counterfactual generator produces semantically consistent substitutes.

In single-modality settings, CARE exhibits the strongest robustness. On MIntRec2.0, it achieves 38.72% accuracy with vision alone (vs. 36.96% for MVCL-DAF) and 39.12% with audio alone (vs. 37.06%). When only text is available, recall improves by more than 13 points. These outcomes indicate that CARE enables intervention-invariant and generalizable intent recognition even under severe information loss. To further examine resilience, we performed a stress test by injecting Gaussian noise into the features (see Appendix B.3).

Table 5: Robustness evaluation under different modality-missing scenarios on the MIntRec and MIntRec2.0 datasets. CARE is compared against the strong baseline MVCL-DAF across various modality combinations. The values in parentheses ($\Delta$ ↑) indicate the absolute performance gain of CARE over MVCL-DAF under the corresponding setting.

| modality combination | | MIntRec | | | | MIntRec2.0 | | | |
|---|---|---|---|---|---|---|---|---|---|
| | | ACC | WF1 | WP | R | ACC | WF1 | WP | R |
| MVCL-DAF | V+A+T | 74.72 | 74.61 | 75.07 | 71.94 | 57.80 | 55.05 | 55.82 | 42.03 |
| | V+A | 43.25 | 42.90 | 43.65 | 40.82 | 46.91 | 46.37 | 46.75 | 42.38 |
| | V+T | 74.85 | 74.50 | 75.12 | 72.44 | 58.12 | 57.85 | 58.06 | 52.41 |
| | A+T | 71.42 | 71.05 | 72.01 | 70.11 | 57.93 | 57.52 | 57.74 | 51.86 |
| | V | 18.00 | 16.95 | 16.85 | 12.68 | 36.96 | 20.18 | 14.64 | 3.26 |
| | A | 25.39 | 22.80 | 23.47 | 18.15 | 37.06 | 20.04 | 13.73 | 3.23 |
| | T | 72.13 | 71.80 | 72.50 | 56.73 | 53.59 | 53.59 | 54.89 | 40.30 |
| our CARE | V+A+T | 76.63 (1.91↑) | 76.56 (1.95↑) | 77.28 (2.21↑) | 74.89 (2.95↑) | 59.86 (2.06↑) | 59.46 (4.41↑) | 59.59 (3.77↑) | 54.06 (12.03↑) |
| | V+A | 45.12 (1.87↑) | 44.85 (1.95↑) | 45.30 (1.65↑) | 42.15 (1.33↑) | 48.33 (1.42↑) | 47.90 (1.53↑) | 48.15 (1.40↑) | 45.22 (2.84↑) |
| | V+T | 75.96 (1.11↑) | 75.68 (1.18↑) | 76.07 (0.95↑) | 73.83 (1.39↑) | 59.27 (1.15↑) | 59.11 (1.26↑) | 59.35 (1.29↑) | 54.02 (1.61↑) |
| | A+T | 72.81 (1.39↑) | 72.53 (1.48↑) | 73.38 (1.37↑) | 72.00 (1.89↑) | 59.32 (1.39↑) | 58.99 (1.47↑) | 59.18 (1.44↑) | 53.71 (1.85↑) |
| | V | 19.64 (1.64↑) | 18.53 (1.58↑) | 18.71 (1.86↑) | 16.03 (3.35↑) | 38.72 (1.76↑) | 21.98 (1.80↑) | 16.92 (2.28↑) | 19.87 (16.61↑) |
| | A | 27.41 (2.02↑) | 24.52 (1.72↑) | 25.11 (1.64↑) | 20.47 (2.32↑) | 39.12 (2.06↑) | 22.45 (2.41↑) | 15.74 (2.01↑) | 18.63 (15.40↑) |
| | T | 73.03 (0.90↑) | 72.80 (1.00↑) | 73.80 (1.30↑) | 72.50 (3.70↑) | 58.93 (2.20↑) | 58.77 (5.18↑) | 59.03 (4.14↑) | 53.50 (13.20↑) |

## 5 CONCLUSION

We presented CARE, a causal learning framework designed to improve robustness in MIR by framing modality completion as a causal intervention and applying adversarial learning to mitigate confounding bias. Experiments on MIntRec and MIntRec2.0 show that CARE achieves state-of-the-art accuracy and maintains robustness under various missing-modality conditions.

The framework's design makes it suitable for real-world applications with incomplete data streams, such as in-car assistants and assistive robotics, where reliability and generalization are critical for human–computer interaction.

This study has several limitations. CARE currently relies on a relatively simple synthesis network and addresses only a single confounder ($M$). Future directions include incorporating stronger generative models for counterfactual construction, expanding the causal graph to capture multiple confounders, and applying the framework to broader multimodal tasks to further assess its generalizability.

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

# A  ALGORITHM AND MODEL DETAILS

## A.1  END-TO-END TRAINING ALGORITHM

Algorithm 1 summarizes the end-to-end training procedure of CARE. For each batch, $G_f$ performs modality completion and fusion to produce $Z_{\text{pooled}}$. This representation is then used to compute (1) the classification loss $\mathcal{L}_{\text{task}}$, (2) the adversarial loss $\mathcal{L}_{\text{adv}}$, and (3) the consistency loss $\mathcal{L}_{\text{consist}}$ for fully observed samples. The combined loss is used to update all model parameters.

---

**Algorithm 1** CARE Framework: End-to-End Training

---

**Require:** Batch $\{X_t, X_a, X_v\}$, modality indicator $M$, label $Y$
**Require:** Models: $G_f, G_d, G_y$
**Ensure:** Updated parameters for $G_f, G_d, G_y$
 1: $Z_{\text{pooled}} \leftarrow G_f(X_t, X_a, X_v, M)$
 2: $\hat{Y} \leftarrow G_y(Z_{\text{pooled}})$
 3: $\mathcal{L}_{\text{task}} \leftarrow \text{CrossEntropy}(\hat{Y}, Y)$
 4: $\mathcal{L}_{\text{adv}} \leftarrow \text{CrossEntropy}(G_d(\text{GRL}(Z_{\text{pooled}})), M)$
 5: **if** $M = 6$ **then**
 6:     Compute $\mathcal{L}_{\text{consist}}$ by comparing predictions under counterfactual missingness
 7: **else**
 8:     $\mathcal{L}_{\text{consist}} \leftarrow 0$
 9: **end if**
10: $\mathcal{L}_{\text{total}} \leftarrow \mathcal{L}_{\text{task}} + \alpha\mathcal{L}_{\text{adv}} + \beta\mathcal{L}_{\text{consist}}$
11: Update parameters of $G_f, G_y, G_d$ using $\mathcal{L}_{\text{total}}$

---

## A.2  DETAILS OF THE MODALITY EXISTENCE INDICATOR ($M$)

As described in Section 3.1, the modality existence indicator $M$ is an integer-valued variable that denotes the presence or absence of each modality for a given sample. Table 6 presents the complete encoding scheme used in this work to represent all possible modality combinations consistently.

Table 6: Encoding of modality existence indicator $M$ for combinations of text ($T$), audio ($A$), and video ($V$) modalities.

| $M$ Encoding Value | Missing Modalities | Present Modalities |
|---|---|---|
| 0 | Text ($T$) | Audio + Video ($A + V$) |
| 1 | Audio ($A$) | Text + Video ($T + V$) |
| 2 | Video ($V$) | Text + Audio ($T + A$) |
| 3 | Text + Audio ($T + A$) | Video ($V$) |
| 4 | Text + Video ($T + V$) | Audio ($A$) |
| 5 | Audio + Video ($A + V$) | Text ($T$) |
| 6 | None | Text + Audio + Video ($T + A + V$) |

## A.3  TWO-STAGE TRAINING PROTOCOL

Our training procedure consists of two sequential phases that together balance model robustness and task-specific performance. The first phase, robustness-oriented pre-training, is conducted for a substantial number of epochs (e.g., 100 epochs) using a relatively high learning rate (e.g., $2 \times 10^{-5}$). During this phase, we introduce significant input uncertainty through modality dropout, where entire modalities are randomly nullified for fully-observed samples with a specific probability. This approach encourages the model to develop representations that are not overly dependent on any single modality, enhancing resilience to incomplete data scenarios and improving generalization.

After completing the pre-training phase, we restore the model parameters that yielded the best validation performance and proceed to performance-oriented fine-tuning for a few additional epochs

(e.g., 5 epochs). In this stage, the learning rate is drastically reduced (e.g., to $1 \times 10^{-7}$) and the modality dropout probability is decreased or disabled. This allows the model to converge more precisely on the task-specific loss surface, refining its discriminative capabilities while retaining the robust foundation established during pre-training. By sequentially combining these phases, the optimization strategy effectively mediates the trade-off between generalization and task-specific performance.

### A.4 EXPERIMENTAL SETUP

All experiments were carried out using the PyTorch framework on a cloud server with a 32GB GPU. For the text modality, we employed the pre-trained BERT model (bert-large-uncased) from the Hugging Face Transformers library. To ensure reproducibility, all models were trained with a fixed random seed.

CARE was evaluated on both the MIntRec and MIntRec2.0 datasets. Key training and optimization hyperparameters were carefully adjusted for each dataset to ensure fair and robust comparisons. While most settings were shared, critical parameters such as learning rate and early stopping patience were tailored to the specific characteristics of each dataset. The detailed configurations used in our experiments are summarized in Table 7.

Table 7: Key training and optimization hyperparameters for the MIntRec and MIntRec2.0 datasets.

| Category | Hyperparameter | Dataset | |
|---|---|---|---|
| | | **MIntRec** | **MIntRec2.0** |
| **Training Parameters** | Number of Epochs | 100 | 100 |
| | Train Batch Size | 16 | 16 |
| | Evaluation Batch Size | 8 | 8 |
| | Test Batch Size | 8 | 8 |
| | Early Stopping Patience | 10 | 8 |
| | Learning Rate (lr) | 2e-5 | 1e-5 |
| **Optimization Parameters** | LR Method | decay | decay |
| | Warmup Proportion | 0.1 | 0.1 |
| | Weight Decay | 0.2 | 0.2 |
| | Gradient Clip | -1.0 | -1.0 |

## B ADDITIONAL ANALYSIS AND RESULTS

### B.1 HYPERPARAMETER ANALYSIS

The CARE framework's training objective involves two primary hyperparameters: $\alpha$, which controls the weight of the adversarial de-confounding loss ($\mathcal{L}_{\text{adv}}$), and $\beta$, which controls the weight of the counterfactual consistency loss ($\mathcal{L}_{\text{consist}}$). To identify optimal values, we performed a systematic grid search and evaluated model performance on both the MIntRec and MIntRec2.0 datasets.

Results are summarized numerically in Table 8 and visualized as 3D performance surfaces in Figure 5. These plots show the WF1 score as a function of $\alpha$ and $\beta$, providing an intuitive view of their interaction.

The analysis indicates that incorporating the adversarial loss ($\alpha > 0$) consistently improves performance compared with the baseline ($\alpha = 0$), which is evident as an upward slope along the $\alpha$-axis in the surface plots. The optimal value is around $\alpha = 0.5$, while increasing it to 1.0 generally leads to a minor decrease in performance. Similarly, increasing $\beta$ from 0 to 1.0 generally enhances performance, as reflected by the surface along the $\beta$-axis, whereas further increasing it to 1.5 tends to slightly reduce performance, suggesting an optimal point for the consistency term. Overall, the best results are obtained when both losses are active, with the peak at $\alpha = 0.5$ and $\beta = 1.0$ clearly visible in the performance surfaces for both datasets, highlighting the complementary contributions of the two components.

Based on these results, we selected $\alpha = 0.5$ and $\beta = 1.0$ as the optimal configuration, which was used in all main experiments reported in this work.

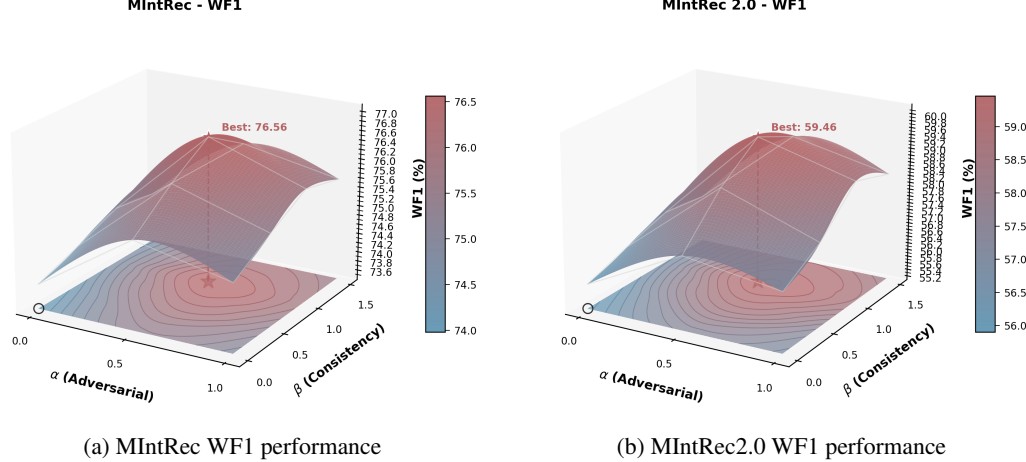

(a) MIntRec WF1 performance          (b) MIntRec2.0 WF1 performance

Figure 5: 3D visualization of WF1 performance as a function of hyperparameters $\alpha$ and $\beta$. The optimal configuration ($\alpha = 0.5, \beta = 1.0$) is marked with a star; the hollow circle at $(0, 0)$ indicates the baseline without either loss.

Table 8: Hyperparameter sensitivity analysis for the loss weights $\alpha$ (adversarial loss) and $\beta$ (consistency loss). The best-performing configuration is highlighted in **bold**.

| $\alpha$ | $\beta$ | **MIntRec** | | **MIntRec2.0** | |
|---|---|---|---|---|---|
| | | **ACC** | **WF1** | **ACC** | **WF1** |
| 0 | 0 | 74.25 | 73.98 | 57.10 | 55.90 |
| 0 | 0.5 | 74.82 | 74.30 | 57.65 | 56.32 |
| 0 | 1.0 | 75.01 | 74.65 | 57.92 | 56.70 |
| 0 | 1.5 | 74.88 | 74.52 | 57.80 | 56.55 |
| 0.5 | 0 | 75.62 | 75.33 | 58.45 | 57.32 |
| 0.5 | 0.5 | 76.05 | 76.02 | 59.05 | 58.60 |
| **0.5** | **1.0** | **76.63** | **76.56** | **59.86** | **59.46** |
| 0.5 | 1.5 | 76.10 | 76.00 | 59.22 | 58.90 |
| 1.0 | 0 | 75.21 | 74.90 | 58.01 | 56.92 |
| 1.0 | 0.5 | 75.68 | 75.40 | 58.55 | 57.60 |
| 1.0 | 1.0 | 76.02 | 75.85 | 59.10 | 58.70 |
| 1.0 | 1.5 | 75.77 | 75.61 | 58.72 | 58.30 |

## B.2 CAUSAL INVARIANCE VISUALIZATION

To qualitatively assess the effectiveness of the adversarial de-confounding mechanism, we visualize the learned intent representation space using t-SNE. Feature representations from both the MIntRec and MIntRec 2.0 test sets are projected into a 2D space, where each point is colored according to its ground-truth intent label and shaped by its feature source (*Real* vs. counterfactual features generated from incomplete modalities such as *Gen_from_AV*, *Gen_from_TV*, and *Gen_from_TA*).

The t-SNE plots reveal that points sharing the same intent label form compact, well-separated clusters in both datasets. This effect is especially evident in MIntRec, where intent categories are balanced, whereas MIntRec 2.0 exhibits greater variability due to long-tailed label distributions. Within each intent cluster, markers representing different feature sources (*Real*, *Gen_from_AV*, *Gen_from_TV*, *Gen_from_TA*) are thoroughly intermixed, and no systematic separation is observed between real and generated features. These observations indicate that the learned representations

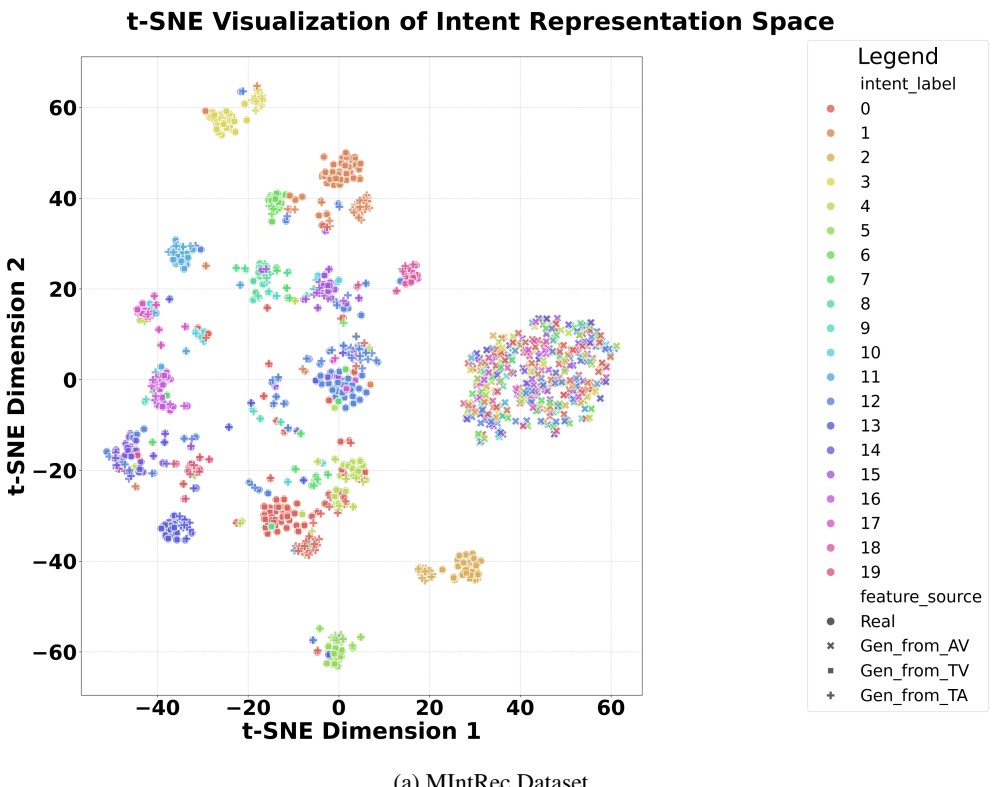

(a) MIntRec Dataset

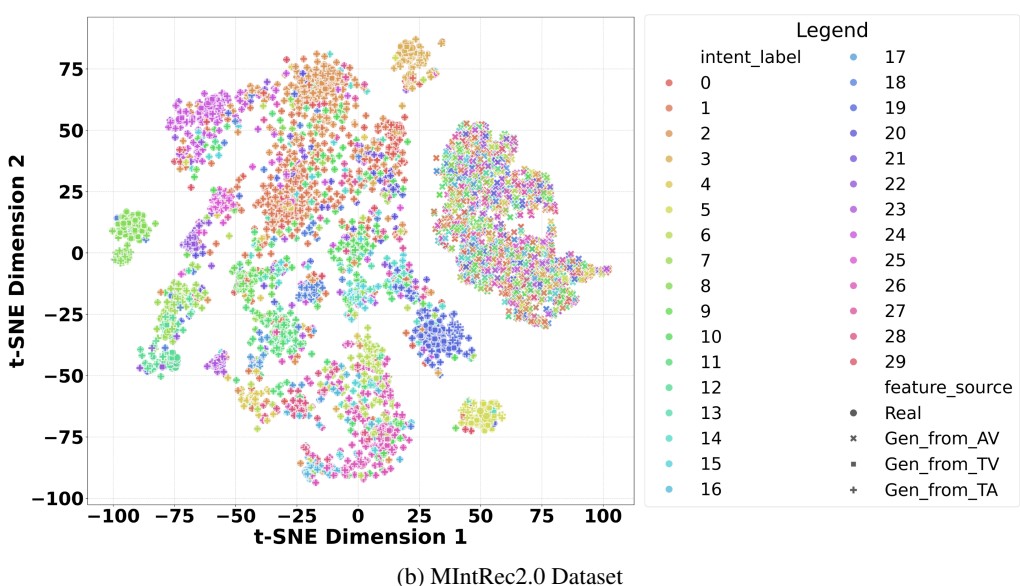

(b) MIntRec2.0 Dataset

Figure 6: t-SNE visualization of the learned intent representation space on (a) MIntRec and (b) MIntRec 2.0 test sets. Each point is colored by intent label and shaped by feature source. Clear clustering by color and strong intermixing of markers demonstrate that the learned representations are discriminative of intent while invariant to modality source.

effectively capture intent-specific information while remaining invariant to the modality source. Overall, the plots demonstrate that CARE produces features that are discriminative across intent categories and robust to variations in modality availability, supporting the framework's goal of reliable multimodal intent recognition under incomplete or corrupted input conditions.

## B.3 ROBUSTNESS STRESS TEST

Table 9: Performance of the CARE model under different levels of Gaussian noise injected into audio and video features. Values in parentheses indicate the performance change ($\Delta$) relative to the no-noise baseline ($\sigma = 0$).

| Noise Level | MIntRec | | | | MIntRec2.0 | | | |
|---|---|---|---|---|---|---|---|---|
| | ACC | WF1 | WP | R | ACC | WF1 | WP | R |
| 0 | **76.63** | **76.56** | **77.28** | **74.89** | **59.86** | **59.46** | **59.59** | **54.06** |
| 0.1 | 75.73 (0.90↓) | 75.48 (1.08↓) | 75.91 (1.37↓) | 73.53 (1.36↓) | 59.27 (0.59↓) | 59.13 (0.33↓) | 59.29 (0.30↓) | 54.57 (0.51↑) |
| 0.2 | 75.51 (1.12↓) | 75.25 (1.31↓) | 75.62 (1.66↓) | 73.32 (1.57↓) | 59.32 (0.54↓) | 59.15 (0.31↓) | 59.28 (0.31↓) | 54.57 (0.51↑) |
| 0.3 | 75.73 (0.90↓) | 75.44 (1.12↓) | 75.78 (1.50↓) | 73.61 (1.28↓) | 59.37 (0.49↓) | 59.22 (0.24↓) | 59.38 (0.21↓) | 54.78 (0.72↑) |
| 0.4 | 75.51 (1.12↓) | 75.28 (1.28↓) | 75.65 (1.63↓) | 73.32 (1.57↓) | 59.37 (0.49↓) | 59.24 (0.22↓) | 59.41 (0.18↓) | 54.61 (0.55↑) |
| 0.5 | 75.51 (1.12↓) | 75.25 (1.31↓) | 75.62 (1.66↓) | 73.32 (1.57↓) | 59.37 (0.49↓) | 59.22 (0.24↓) | 59.39 (0.20↓) | 54.72 (0.66↑) |

To systematically evaluate the performance stability of the CARE framework under non-ideal conditions that simulate real-world data perturbations, we conducted a controlled noise injection stress test. The core hypothesis is that a model learning causally-informed, abstract representations should exhibit greater resilience to noise. The test was performed as a post-training evaluation, where Additive White Gaussian Noise, sampled from $\mathcal{N}(0, \sigma^2)$, was injected into the audio and video features of test samples, with noise intensity $\sigma$ ranging from 0.1 to 0.5.

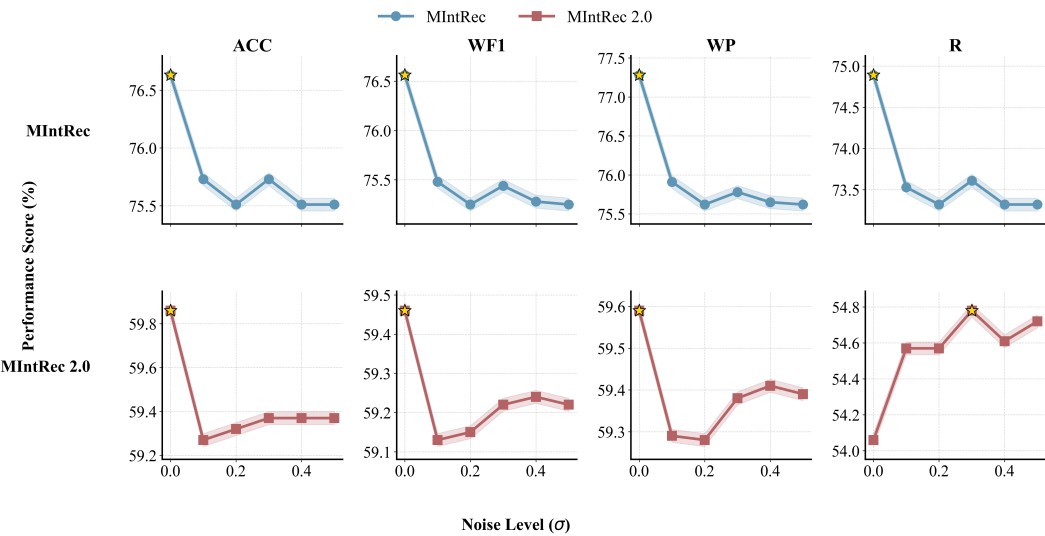

Figure 7: Performance curves of the CARE model on MIntRec and MIntRec2.0 under varying levels of Gaussian noise ($\sigma$). Each column represents a different evaluation metric. The nearly flat trajectories, especially for ACC and WF1, highlight the model's exceptional robustness to noise.

The results, visualized in Figure 7 and detailed numerically in Table 9, indicate that the CARE framework maintains exceptional stability across all noise levels for both datasets. The ACC, WF1, and WP metrics remain nearly constant, and the small numerical drops confirm this visual trend. For example, on the more challenging MIntRec2.0 dataset, the ACC score decreases by at most 0.59% at $\sigma = 0.1$, demonstrating strong resilience to feature corruption and supporting the claim that CARE's learning of abstract causal relations reduces dependence on low-level, noise-sensitive features. An interesting phenomenon is observed in the Recall (R) metric for MIntRec2.0, where performance slightly increases with the introduction of noise. This is not indicative of an error, but likely reflects a subtle shift in the model's precision–recall trade-off, possibly due to a test-time regularization

effect. The injected noise may obscure certain overly confident but misleading features, leading the model to produce slightly less precise but more inclusive predictions. This effect is supported by the data: at $\sigma = 0.3$, Recall increases by 0.72 while Precision decreases by 0.21. Consequently, the WF1 score remains highly stable, effectively balancing both metrics.

Overall, the stress test demonstrates that CARE not only exhibits robust performance under noisy conditions but also maintains a stable and effective balance between precision and recall, even when input features are significantly corrupted.

## C    USE OF LARGE LANGUAGE MODELS

In the preparation of this manuscript, we utilized a Large Language Model (LLM) as an assistive tool. The primary and sole role of the LLM was for language polishing and editing. Specifically, it was used to improve grammar, enhance sentence clarity and flow, and ensure stylistic consistency throughout the paper.

Crucially, the LLM was not used for the generation of core scientific ideas, the design of the CARE framework, the analysis of experimental results, or the formulation of our conclusions. All authors have carefully reviewed and edited the outputs from the LLM and take full responsibility for the final content of this paper, ensuring its scientific accuracy and integrity.

## D    SUPPLEMENTARY MATERIALS: SOURCE CODE

In order to facilitate the replication of our experimental results, we have provided the source code for our method. The code is hosted in an anonymous repository and includes the core code in this paper. The repository can be accessed via the following link:

`https://anonymous.4open.science/r/CARE-99ED-J`

