# OpenReview forum: "CARE: Causal Intervention and Adversarial Learning for Robust Multimodal Intent Recognition"
_ICLR.cc/2026/Conference — ICLR 2026 Conference Withdrawn Submission_

### Official Review · Reviewer_xoYK · 2025-10-31

**Soundness:** 2
**Presentation:** 3
**Contribution:** 2
**Rating:** 4
**Confidence:** 4

**Summary:**

The paper proposes CARE for multimodal intent recognition (MIR). CARE combines (1) a counterfactual generation module that synthesizes missing-modality features, (2) adversarial de-confounding via a gradient-reversal discriminator to remove dependence on the modality-availability indicator, and (3) a dynamic attention fusion block. Experiments on MIntRec and MIntRec 2.0 report gains (notably in recall on MIntRec 2.0), ablations for each component, and robustness to synthetic noise.

**Strengths:**

+ Clear problem framing (robust MIR under missing modalities).
+ Figures, algorithms, and training details are generally clear.
+ Ablations indicate each component contributes to the final performance.

**Weaknesses:**

- The evaluation is missing simpler baselines that would help justify the framework's complexity. While CARE is compared to strong MIR methods, it is unclear if the complex counterfactual generator is necessary. A simpler baseline, such as using modality dropout or a basic imputation method combined with the DAF and adversarial de-confounding, would be crucial for demonstrating the specific value of the generation module.
- The generator seeds plus projection networks create plausible substitutes, but the paper does not verify whether generated features preserve label-conditional structure (e.g., calibration of class-conditional likelihoods, cycle-consistency, or causal faithfulness checks). This weakens the central claim that CARE captures “intervention-invariant” semantics.
- Reported improvements (often ~1–2% on MIntRec; larger recall gains on MIntRec 2.0) lack variance estimates or statistical tests.
- Very minor: The image source in Figure 1 is captioned as "The Big Bang Theory", but the character seems to be from the sitcom "Friends."

**Questions:**

(1) Does the counterfactual generator require complete, all-modality samples to learn its "what-if" synthesis? If so, how would the method be trained on datasets where certain modality combinations are entirely absent?

(2) The adversarial de-confounding relies on a discriminator for 7 modality combinations. How is this approach expected to scale to scenarios with more than 3 modalities (e.g., 5 or 10), where the number of combinations ($2^N - 1$) would grow exponentially?

(3) What specific design choices or assumptions make the generated features "counterfactual" (implying a causal what-if scenario) rather than simply "imputational" (a statistical best-guess for missing data)?

(4) How sensitive are results to generator design (seed size, architecture) and GRL weight α/β across seeds?

(5) The added components (generator, discriminator) seem likely to increase the model's size and inference time. Could you provide the inference-time cost (e.g., latency or GFLOPs) and total parameter count relative to the baselines?

---

### Official Review · Reviewer_yNqJ · 2025-10-31

**Soundness:** 2
**Presentation:** 3
**Contribution:** 3
**Rating:** 4
**Confidence:** 3

**Summary:**

The paper proposes CARE for Multimodal Intent Recognition (MIR) with missing modalities. CARE combines:
- (i) a counterfactual generation module that synthesizes features for absent modalities using modality-specific projections plus a learnable seed tensor, and
- (ii) adversarial de-confounding via a gradient reversal layer (GRL) to make the pooled representation invariant to the modality-combination indicator M.

A Dynamic Attention Allocation Fusion (DAF) module integrates observed and generated features; the total loss adds task cross-entropy, adversarial loss, and a counterfactual consistency KL term computed by masking one modality on fully observed samples. Experiments on MIntRec and MIntRec2.0 report SOTA results.

**Strengths:**

- **Clear causal framing and implementation**. The paper formalizes M (which marks which modalities are present or generated) as a confounder and enforces $Y \perp M \mid Z_{\text{pooled}}$ via GRL—grounded in domain-adversarial training (DANN). The causal graph and the invariance objective are explicit, and the two-stage training (robustness pre-train then fine-tune) is well described.
- **Simple yet effective counterfactual generator**. The generation module is lightweight but yields the largest gains in ablations.
- **Stress tests and missing-modality matrix**. Cross-scenario results (all 7 modality configurations) and Gaussian-noise stress tests support the robustness claim; the per-component ablations are informative.

**Weaknesses:**

- **Causal assumption on $M$ may be over-strong**. CARE treats $M$ as a confounder. I wonder if this holds true in every circumstance. E.g. in MIR, missingness can be informative (e.g., some intents surface mainly via audio/vision cues). Forcing invariance to $M$ may suppress predictive signals or induce negative transfer; the paper lacks diagnostics (e.g., mutual information $I(Z,M)$, or performance when $M$ is truly informative).
- **Limited set of baselines for robustness**. Besides four MIR baselines, the study omits invariance methods (e.g., IRM, GroupDRO) and missing-modality adaptation approaches that could be competitive under incomplete inputs. Including one such baseline would strengthen claims of causal robustness (e.g. [1]).
- **Scope of datasets**. Both benchmarks are from the same family (E.g. MIntRec, MIntRec2.0). This is somehow restrictive. Broader evaluation (e.g., other multimodal dialogue datasets) would better evidence generalization.
- **Fairness/implementation details**. The Authors involve set of dedicated unimodal encoders -- E.g. text uses BERT-large fine-tuning, while video is a custom transformer, ... It’s unclear whether baselines were matched for encoder capacity and pretraining, leading to potential bias in selecting these as encoders.

[1] Arjovsky, Martin, et al. "Invariant risk minimization." arXiv preprint arXiv:1907.02893 (2019).

**Questions:**

Please refer to Weaknesses.

---

### Official Review · Reviewer_577P · 2025-11-01

**Soundness:** 1
**Presentation:** 2
**Contribution:** 1
**Rating:** 0
**Confidence:** 4

**Summary:**

The paper attempt to do robust multimodal intent recognition by applying causal intervention. The basic causal model in the draft is that the modality availability variables causes the label $Y$ as well as the pooled feature $Z_{\text{pool}}$, and consider the modality availability variable as a confounder. From causality's perspective, I highly doubt the proposed model. First of all, confounder should be a latent variable in the context of causality, while the modality availability variable, I guess, should be observable. Second, how can we justify that modality availability variable is a cause of true label Y?

There is a so-called counter-factual module in the overall model. Is that really counter-factual? How do you recover the latent variable and then do the intervention on the latent variable to achieve counter-factual? How do you ensure the consistency of the recovered latent variable against the true latent variable?

**Strengths:**

I believe the causal model in the paper should be considered as ``not even wrong''.

**Weaknesses:**

The basic model needs to be justified and it is highly likely to be wrong. Thus the paper is not really doing any intervention or counter-factual using a causal model.

For example, in Section 3.4, the authors claims that they are doing counterfactual, and they are generating features for missing modality. In the so-called ``causal`` graph, the authors thinks that modality will have effects on the $Z_{pool}$. Thus for the counterfactual, the authors should try to solve the problem of ''what if we have observed the modality, how would the feature $Z_{pool}$ and label $Y$ like''. Thus if the ``causal`` model in the draft is correct, then observing a modality or not would affect the ground truth label $Y$, and thus it should also affect all the observations in all other modality, otherwise how can the modality availability affect the ground truth label $Y$? The authors just use a generative model to generate features for the missing label, how do you ensure the generated feature align with other observed feature? How do the generated feature affect the ground truth label?

I guess the $Y$ in the ``causal'' model is not the ground truth $Y$, instead it is the predicted $Y$ from the model, which represents the model’s internal belief or approximation of the outcome under the assumed causal mechanism rather than the true data-generating process. In this case, I can hardly view it as a causal model.

**Questions:**

See above summary and weakness.

---

### Note · Authors · 2025-11-12

I have read and agree with the venue's withdrawal policy on behalf of myself and my co-authors.